



# Flexible approach for quantifying average long-term changes and seasonal cycles of tropospheric trace species

David D. Parrish[1,2], Richard G. Derwent[3], Simon O'Doherty[4], and Peter G. Simmonds[4]

[1] Institute for Environmental and Climate Research, Jinan University, Guangzhou, China
[2] David.D.Parrish, LLC, Boulder, Colorado, USA
[3] rdscientific, Newbury, Berkshire, RG14 6LH, United Kingdom
[4] Schoo1 of Chemistry, University of Bristol, Cantocks Close, Bristol, BS8 1TS, United Kingdom

*Correspondence to*: David D. Parrish (david.d.parrish.llc@gmail.com)

**Abstract.** We present an approach to derive a systematic mathematical representation of the statistically significant features of the average long-term changes and seasonal cycle of concentrations of trace tropospheric species. The results for two illustrative data sets (time series of baseline concentrations of ozone and $N_2O$ at Mace Head, Ireland) indicate that a limited set of seven or eight parameter values provides this mathematical representation for both example species. This method utilizes a power series expansion to extract more information regarding the long-term changes than can be provided by oft-employed linear trend analyses. In contrast, the quantification of average seasonal cycles utilizes a Fourier series analysis that provides less detailed seasonal cycles than are sometimes represented as twelve monthly means; including that many parameters in the seasonal cycle representation is not usually statistically justified, and thereby adds unnecessary "noise" to the representation and prevents a clear analysis of the statistical uncertainty of the results. The approach presented here is intended to maximize the statistically significant information extracted from analyses of time series of concentrations of tropospheric species regarding their mean long-term changes and seasonal cycles, including non-linear aspects of the long-term trends. Additional implications, advantages and limitations of this approach are discussed.

## 1 Introduction

Utilizing observations to fully characterize the four-dimensional (latitude, longitude, altitude and time) concentration distribution of a trace tropospheric species is a daunting prospect. This paper discusses an analysis approach for quantification of only a small part of the full distribution - the mean long-term changes and seasonal cycle at a particular point in the troposphere - but it provides that quantification in an accurate, precise and simple form. The discussion focuses on monthly mean, baseline ozone (Derwent et al., 2018a) and $N_2O$ concentrations reported for the surface site at Mace Head, Ireland, but the analysis approach is general, and so can be applied to other locations and trace species. This approach extends the techniques developed in earlier publications: Parrish et al. (2012; 2014; 2017) for long-term changes and Parrish et al. (2016) and Derwent et al., (2016, 2018b) for seasonal cycles. This extension provides consistently defined parameters with confidence limits that quantify these systematic temporal variations.





From a wider, more formal perspective, Bowdalo et al. (2016) discuss the temporal variability of hourly average, tropospheric ozone concentrations through an extensive spectral analysis. They identify two distinct scaling regimes of ozone variability, one at high frequencies with periods from 2 hours to about 10 days, and a second at lower frequencies with 10 day to 5 year periods (the maximum period they considered). In analogy with spectral analysis of meteorological

variability, Bowdalo et al. (2016) identify the higher frequencies as the "weather" regime, driven by meteorological processes with frequencies up to those of planetary-scale weather systems. Meteorological frequencies with periods greater than ~10 days are driven by the average of the largest planetary-scale weather systems, which they term "macroweather". They also suggest that there would be a third "climate" regime beginning between 10 and 100 years caused by low-frequency interactions such as solar, volcanic, or anthropogenic forcings. Since they limited their consideration to time series

of 5 years, they could not identify any evidence of the climate regime. In this work we consider some of the longest available observational records (as long as thirty years), but work with monthly mean concentrations. Thus, we aim to characterize the macroweather regime and the higher frequency fraction of the climate regime.

The focus of our analysis is on mean seasonal cycles and long-term changes spanning the complete data records. By considering monthly average data we avoid most of the influence from higher frequency variability, although interannual

variability of lower frequency weather regime phenomena contributes "noise" to the monthly averages, and thereby affects the precision of the mean seasonal cycle and long-term change quantifications. Even when using monthly averaged data, observable autocorrelation remains in the data after accounting for the seasonal cycle and long-term changes. This autocorrelation is estimated from an autoregressive process, and accounting for the autocorrelation results in expanded error estimates for the derived parameters.

An accurate and precise quantification of the mean long-term changes and seasonal cycles of trace species distributions with well-defined confidence limits is the ultimate goal of the analysis. This quantification meets three needs: 1) it provides a robust, minimum set of parameters capturing the statistically significant information in the observational data set regarding these two temporal variations, parameters that can serve as metrics for quantitative comparisons of long-term changes and seasonal cycles between data sets collected at different locations or for different species; 2) these same metrics serve as a

basis for evaluation of results from models that simulate these temporal features of atmospheric concentrations; and 3) it provides a coherent, conceptual view of these features of the species' concentration distribution, a view that can provide indications of the need for more detailed studies of particular aspects of the distribution.

Importantly, no physical model underlies the statistical analysis. Instead we use two mathematical series to fit the long-term change (a power series) and the seasonal cycle (a Fourier series). These series provide flexible fits to these temporal

variations, even though their functional form is not known a prioi. The data sets themselves dictate the functional form defined by the mathematical series. To avoid over fitting the data, the number of terms in each series is limited to only those that are statistically significant. Without an underlying model, care must be exercised in the physical interpretation of the derived parameter values.



## 2 Example Data Sets

The analysis approach discussed here is exemplified through application to two data sets collected at Mace Head Ireland: monthly mean concentrations of ozone ($O_3$) and nitrous oxide ($N_2O$) filtered for baseline conditions. These two data sets provide an informative contrast – ozone has a strong seasonal cycle with a relatively small long-term change, while $N_2O$ has

a relatively small seasonal cycle superimposed on a pronounced long-term change. Derwent et al. (2018a) fully describe the ozone data set; it covers the 30-year period from April 1987-April 2017. The $N_2O$ data were provided by the AGAGE program and downloaded from the public U.S. Department of Energy (DOE) Carbon Dioxide Information Analysis Center (CDIAC) website (http://cdiac.esd.ornl.gov/ndps/alegage.html); these data cover the 22-year period from March 1994-September 2016. All AGAGE data are available from the public AGAGE website (http://agage.mit.edu/data), and the World

Data Center for Greenhouse Gases (WDCGG) in Japan (http://gaw.kishou.go.jp/wdcgg), as well as on the CDIAC website.

## 3 Analysis Approach

The overall goal of the analysis is to quantify the minimum set of parameters (including robust confidence limits) that mathematically describes the mean long-term evolution and seasonal cycle of an atmospheric trace species' concentrations, within the limits of statistical significance, from a time series of measured concentrations. The minimum set of parameters is

desired because 1) it minimizes the possibility of over fitting to the data and 2) the analysis can quantify these parameters to the highest precision, which thus provides the most concise picture of the variations and the most stringent metrics for comparisons of different data sets and for model evaluation through model-measurement comparisons. In Sections 3.1-3.4 the analysis is illustrated through application to Mace Head baseline ozone observations (Derwent et al., 2018a), but the method is generally applicable to other trace species; Section 3.5 illustrates the application to Mace Head baseline $N_2O$

observations.

Here we first quantify the long-term changes through a power series fit to the time series of monthly mean ozone concentrations (Section 3.1). The derived function defining the long-term changes is then used to detrend the monthly mean data (Section 3.2) in order to facilitate characterization of the seasonal cycle through a Fourier transform and harmonic analysis of the detrended monthly means (Section 3.3). Finally, a non-linear regression fit of a function containing the

statistically significant terms of both the power and Fourier series gives the most precise determination (i.e., yields the smallest confidence limits) of the derived parameters (Section 3.4). Sections 3.6 through 3.8 discuss additional statistical features of the data, including their influence on the confidence limits of the derived parameter values.

Working with monthly mean data reduces the impact of autocorrelation due to weather regimes and results in residuals that are more Gaussian in nature. Restricting the analysis to monthly mean time data rather than working with higher frequency

data does not reduce the statistically significant information regarding the average long-term trends or seasonal cycles. A qualitative explanation for this can be given. Deriving monthly means from higher frequency data (e.g., hourly or daily mean





data) is an averaging process that minimizes the sum of the squares of the deviations of the higher frequency data from the derived monthly means. The fitting procedures employed in the analysis here minimize the sum of the squares of the deviations of the monthly mean data from the derived long-term changes and seasonal cycles. The overall result is independent of whether the sum of the squares of the deviations is minimized in two steps (monthly mean calculation

followed by further fits), or in one step (extracting long-term trends and seasonal cycles directly from the higher frequency data). One example of this independence is shown in Section 3.1 where a fit of the long-term change function to annual mean data gives results equivalent to the fit to monthly mean data. We work with monthly mean data because they provide clearer illustrations of the method and its results, compared to higher frequency data.

### 3.1 Long-term Change Analysis

A power series fit is a general and convenient means to quantify the long-term temporal evolution of a time series of concentration measurements. This is a general approach in that no underlying assumptions are made regarding the functional form of the temporal evolution of the data set, since any continuously varying curve can be fit to any desired accuracy given enough terms in a power series. In practice, the power series fit is obtained through a non-linear regression fit of monthly mean data to a polynomial as indicated in Equation 1:

$$[O_3] = a + bt + ct^2 + dt^3 + \dots \tag{1}$$

The fits utilized in this work include all terms in Eq. 1 whose coefficients are statistically significant at the 95% confidence level. This means that as longer data records develop, additional terms can be added and new insights can be gained. Figure 1 shows a fit (black solid curve) to monthly mean baseline ozone data (blue solid circles) obtained at Mace Head Ireland (Derwent et al., 2018a). The annotation gives the derived values (with 95% confidence limits) for the first three coefficients

of Eq. 1. The fit to the calendar annual mean data (violet dotted line and larger violet symbols) are also shown. Table 1 compares the coefficients derived from these fits. For this data set, only the first three terms of Eq. 1 are retained, as the coefficients of higher order terms are not statistically significantly different from zero (see derived $d$ parameters in Table 1). All three parameter values derived from the two fits agree within their confidence limits; the small differences are due to the exclusion of the partial years of data at the beginning and end of the data record when calculating the calendar means. The

scatter of the annual means about the fitted curve is considerably smaller than that of the monthly means (compare root-mean-square deviation, RMSD, values in Table 1) since the variability associated with the seasonal cycle has been removed by the annual averaging period.

To more precisely determine the coefficients, it is important for the time origin to be well within the time span of all the data series considered. Here we choose year 2000 (i.e., t in Eq. 1 equals year-2000). If the time origin is selected outside the time

spanned by the data (an extreme example would be year 0), the confidence limits of the derived parameters and the absolute values of $a$ and $b$ (but not $c$) change, but the fitted curve does not change. With year 2000 chosen as the time origin, the first coefficient ($a$, with units ppb $O_3$) is the intercept of the fitted curve at the year 2000; it quantifies the absolute magnitude of





the average ozone concentration at that year. The second coefficient ($b$, with units ppb $O_3$ $yr^{-1}$) is the slope of the fitted curve at that same year; it gives the best estimate of the (continually varying) time rate of change of ozone at that particular time. Finally, the third coefficient ($c$, with units ppb $O_3$ $yr^{-2}$) is equal to one-half of the (constant) time rate of change of the slope of the fitted curve. This third term is important for characterizing the non-linear aspects of long-term behavior of the data.

Many published studies rely on various approaches to analyze long-term trends through linear fits; the recent Tropospheric Ozone Assessment Report (TOAR) project (Chang et al., 2017; Gaudel et al., 2018; Lefohn et al., 2018) takes this approach. The focus of TOAR is on shorter measurement records at hundreds of sites where only the first two terms of Eq. 1 are statistically significant, so their choice is appropriate. Linear trend approaches can accurately quantify the average rate of change in concentrations over a measurement record of any length, but do not fully quantify the long-term temporal

evolution of data sets with strong non-linear behavior, as is the case in the Mace Head ozone data illustrated in Figure 1. The choice of examining linear behavior or more complex modes of change depends upon the purpose of the analysis; this study focuses on deriving scientific insights into concentration changes that may be driven by non-linear factors.

The long-term fit to the Mace Head data finds a statistically significant, negative value for c, with ozone concentrations increasing early in the data record, reaching a maximum, and then decreasing later in the record. When three terms are

included, Equation 2 allows the calculation of the year that the maximum was reached, $year_{max}$:

$$year_{max} = -b/2c + 2000. \qquad (2)$$

The $year_{max}$ calculated from Eq. 2 is included in Table 1, which is within the time period of the observational record. Extrapolation to a maximum year outside the observational time period would depend, in part, on the scientific understanding of the data, including objective indications that the driving factors will result in a maximum under the existing

conditions.

In summary, only three parameters are required to describe the long-term changes in the Mace Head ozone data set; additional terms are not statistically significant, and are therefore omitted from the analysis. This parameter set is $a$, $b$ and $c$, or equivalently $a$, $c$ and $year_{max}$, with the last derived from Eq. 2. The second parameter set has more direct physical significance for this time series. Derwent et al. (2018a) conducted a similar analysis of the long-term change in this data set

and obtained statistically equivalent results.

**3.2 Detrending monthly mean data**

The time series of monthly mean ozone data can be detrended simply by subtracting the second and third terms of the fit to Eq. 1 from the original time series; Figure 2 illustrates the results. As expected, no significant long-term change remains; the average of the detrended data agrees with the $a$ parameter derived above (i.e., the year 2000 intercept of the original fit); and

their standard deviation agrees with the RMSD of the original monthly means about the long-term trend fit to Eq. 1. All 3 statistically significant terms of Eq. 1 could be subtracted from the monthly means, which would give detrended data





averaging zero with the same standard deviation; subtracting only the second and third terms preserves the year 2000 intercept as the mean of the data set.

**3.3 Seasonal Cycle Analysis**

The quantitative analysis of the seasonal cycle has two steps; first, a Fourier analysis determines the number of statistically

significant harmonic contributors to the seasonal cycle of the detrended data, and second, a least-squares fit of those data to the significant harmonic terms provides a set of parameters that quantify the seasonal cycle to the fullest extent that is statistically justified.

A Fourier Transform of a time series of data such as illustrated in Figure 2 captures the information of that time series in frequency space, i.e., as a series of sine functions whose magnitude and phase are expressed as a sequence of complex

numbers. Plotted in Figure 3 are results from the Fourier transform of the data in Figure 2. These are the magnitudes of the real parts of each term, normalized to give the magnitude of the respective sine functions plotted as a function of frequency. There is one point that is off scale at zero frequency, which gives the magnitude of the average of the detrended monthly means. The fundamental (frequency = 1 yr$^{-1}$) and the second harmonic (frequency = 2 yr$^{-1}$) terms clearly have much greater magnitudes than any of the other terms of non-zero frequency. Terms of frequencies < 1 yr$^{-1}$ describe the systematic, multi-

15   year variability that contributes to deviations from a purely repetitive seasonal cycle in Figure 2. There is an indication that the third harmonic (frequency = 3 yr$^{-1}$) may have a significant magnitude, but it is on the edge of statistical significance; in the following analysis, only the fundamental and second harmonic terms will be considered further. This approach is consistent with that of Parrish et al. (2016), who found that, at most, two terms were required to quantify the seasonal cycle of monthly mean ozone concentrations in the marine boundary layer and in the lower free troposphere.

The Fourier transform indicates that the seasonal cycle of the detrended data is quantitatively described by two terms – the fundamental and the second harmonic – plus a third constant term equal to the annual average. The second step in this analysis is to fit these three terms to the detrended monthly means through a least-squares regression to Equation 3:

$$[O_3] = y_o + A_1*\sin(\chi + \phi_1) + A_2*\sin(2*\chi + \phi_2). \quad\quad (3)$$

Figure 4 illustrates this fit for the detrended data of Figure 2. The second and third terms in Eq. 3 are the fundamental and

second harmonic. If the Fourier transform indicated one or more additional harmonics terms were statistically significant, an additional term would be added to Eq. 3 for each additional harmonic, but for the data sets investigated here, no additional harmonics are statistically significant. Two parameters, the amplitude, $A$, and the phase angle, $\phi$, are required to define each of these sine functions. $y_o$ is the annual average ozone concentration over the entire data set, and from the discussion above, must equal both $a$ (the year 2000 intercept derived from the fit to Eq. 1) and the average of the detrended data illustrated in

Figure 2. In Eq. 3 the variable $\chi$ spans one year's time period in radians from 0 to $2\pi$. The parameters derived from the least-squares fit are annotated in Figure 4; they agree closely with those derived for Mace Head by Parrish et al. (2016) (see their





Table 2). The small differences between the results here and in that earlier work are due to the baseline reanalysis and extra years of measurements (Derwent et al., 2018a) now available from Mace Head. The extra years of measurements have resulted in noticeably smaller confidence limits for most of the derived parameters. Derwent et al. (2018a) conducted a similar analysis of this seasonal cycle and obtained statistically equivalent results.

**3.4 Improved Confidence Limits through Simultaneous Long-term Change and Seasonal Cycle Analysis**

It is possible (and preferable) to do a simultaneous fit to the long-term change and seasonal cycle by utilizing an iterative, non-linear regression to Equation 4, which combines the first three (or four) terms of Eq. 1 with the final two terms of Eq. 3, giving a total of seven (or eight) parameter values:

$$[O_3] = a + bt + ct^2\ (+ dt^3) + A_1 * \sin(\chi + \phi_1) + A_2 * \sin(2 * \chi + \phi_2) + residuals, \qquad (4)$$

where *residuals* represent the unexplained portion of the data, and will be examined in Section 3.6. Whether the fourth term (or even additional terms) of Eq. 1 are included in Eq. 4 depends upon whether each additional term is statistically significant. The violet curve in Figure 5 shows the fit of Eq. 4, and the values derived for the seven parameters are annotated. (For ozone the fourth terms in Equations 1 and 4 are not statistically significant, so no value is given for the *d* parameter.) The results here are nearly identical to those discussed earlier, except the confidence limits for the *a, b, c* parameters are

smaller than those derived in Figure 1 (see Table 2). This improvement is due to simultaneously treating the two systematic sources of data variability (i.e., the long-term change and the seasonal cycle).

**3.5 Analysis of Nitrous Oxide Time Series**

The preceding sections developed and illustrated the application of Eqs. 1 through 4 for a time series of monthly mean ozone concentration data, but in principle these equations and analysis approach can be applied to a series of measurements of any

trace species. For example, Figure 6 illustrates the analogous analysis of the time series of monthly mean, baseline-selected nitrous oxide ($N_2O$) measurements from Mace Head. This time series (Figure 6a) is significantly different from that of ozone, with the long-term change dominating the variability of the data, perturbed by only a relatively small seasonal cycle. Further, with ozone only three terms of Eq. 1 are statistically significant, but for $N_2O$ the fourth (cubic) term is also statistically significant (while higher order terms are not). Figure 6a shows fits of Eq.1 for two, three and four terms; it is

difficult to discern the differences between these fits in the figure, but as the annotations indicate, these differences are statistically significant. For $N_2O$ the quadratic term in the three-term fit is positive, indicating that the rate of increase of nitrous oxide has, on average, accelerated over the measurement record in contrast to ozone whose rate of increase decelerated. The statistically significant cubic term shows that the acceleration of the rate of increase has not been constant over the measurement record; Section 3.7 discusses these issues in more detail.

The $N_2O$ data can be detrended as for ozone by subtracting the second, third and fourth terms of the fit to Eq. 1 from the time series (results not shown). The Fourier transform of the detrended data (Figure 6b) is similar to that of ozone in that the only



important harmonic terms are the fundamental and second harmonic. For comparison the Fourier Transform results are shown for data detrended with both the cubic and quadratic fits; the magnitudes at frequencies $< 1$ yr$^{-1}$ are significantly smaller for the cubic fit compared to the quadratic fit, reflecting the reduced variability of the monthly mean data about the cubic fit. Also, for N$_2$O the magnitudes at frequencies $< 1$ yr$^{-1}$ are relatively larger than those for ozone (Figure 3). These

larger magnitudes reflect the noticeable interannual departures of the N$_2$O monthly means from the fitted curve (violet) in Figure 6d. For example, in the years near 2000, the data appear to be significantly smaller than the fit before 2000, and higher after that year. Investigating statistically significant departures, such as this example, may yield additional information regarding sources or sinks of N$_2$O (or other trace species investigated through this analysis approach).

The fit of the detrended data to Eq. 3 to define the seasonal cycle (Figure 6c) is also similar to that of ozone in that the

phases of the two harmonics are similar (see Table 2) for these two species, agreeing (or nearly agreeing) within their confidence limits. This close correspondence is consistent with the long-standing observation that many trace gases show a springtime maximum and a summertime minimum at Mace Head (e.g., Derwent et al., 1998); the cause(s) of this correspondence is an issue warranting further investigation. Finally, Figure 6d illustrates the fit of the original data (plotted in Figure 6a) to both the long-term change and the seasonal cycle as defined by Eq. 4 with the inclusion of the cubic term

from Eq. 1.

## 3.6 Autocorrelation and Parameter Confidence Limits

Through the preceding discussion the statistical fitting ignored any autocorrelation in the data. Systematic intra- or inter-annual variability associated with persistent meteorological and/or climate variability, could possibly cause autocorrelation in these data sets. If such autocorrelation is significant, we expect that the derived parameter values would not be

significantly affected, but the confidence limits derived for those parameter values would be unrealistically small. Parrish et al. (2016) considered this issue for ozone data sets from several sites within the marine boundary layer throughout the globe, and found it to have only small influence. Here we discuss this issue from a more general perspective, and illustrate this discussion through the two example Mace Head data sets.

The time series of the residuals (i.e., the deviations between the monthly mean baseline ozone concentrations and the fit of

Eq. 4 to these means) for the two example data sets are shown in Figures 7a and 8a. (For N$_2$O the residuals are shown for both the quadratic and cubic fits to the long-term change.) The characteristics of these time series differ noticeably; the N$_2$O residuals (Figure 8a) show much more coherent variability than is apparent in the more chaotic ozone residuals (Figure 7a).

The autocorrelations of the time series are shown in Figures 7b and 8b. Each plot shows the correlation of the time series of the monthly means with a duplicate of itself as a function of a time offset (i.e., month lag) between the time series and its

duplicate. When the lag is zero, the correlation is perfect (i.e., autocorrelation coefficient = 1), and as the lag increases, the autocorrelation coefficient decreases. These plots differ markedly between the two species. As expected, the more chaotic time series of the ozone residuals shows smaller autocorrelation coefficients (Figure 7b), decreasing in an approximately



exponential manner with a time constant, tau, ≈ 1 month as the time offset increases. The autocorrelation of the $N_2O$ residuals (Figure 8b) is greater, also decreasing in an approximately exponential manner with tau ≈ 4 and 10 months for the cubic and quadratic fits, respectively). Leith (1973) discusses the degree to which autocorrelation affects the confidence limits of parameters derived from observational time series, and finds that the confidence limits increase proportionally to $(2 tau)^{1/2}$. Thus for the two example data sets discussed here, the confidence limits annotated in Figures 1 and 4-6 and included in Table 1 must be increased by factors of 1.4 and 4.2 for ozone and $N_2O$ (for cubic long-term change fit), respectively; Table 2 gives the corrected confidence limits for the final values derived for the seven or eight parameters.

As is common to all basic treatments of error propagation, the confidence limit analysis presented here is based on the assumption that the residuals of the fits are Guassian distributed. The nearly linear relationships in Figures 7c and 8c (at least for the cubic long-term change fit) give a qualitative indication that this assumption is approximately valid. Each time series has a few apparent outliers of unknown cause; since the prevalence of these outliers is small (no more than 1 to 2%), and they are not greatly outside the general distribution, their influence is believed to be minor, and will not be considered further.

It is notable that Figure 8 clearly reflects the improvement made by the addition of the cubic term to the long-term change fit for the $N_2O$ time series. The standard deviation of the residuals is reduced, the degree of autocorrelation is reduced, which results in improved confidence limits for all of the derived parameters, and the residuals are more closely fir a Gaussian distribution.

**3.7 Rate of Change of Concentrations**

Estimates of the rate of change of the mean concentrations of ozone (or other species) can be derived through differentiation of Eq. 1 to give Equation 5,

$$d[O_3]/dt = b + 2ct \ (+ \ 3dt^2), \qquad (5)$$

where the third term applies when a cubic fit to the long-term change is statistically justified. The quadratic fits to the two species indicate that over their respective data records, the rate of increase on average decelerated at ~0.04 ppb $yr^{-2}$ for ozone and accelerated at ~0.10 ppb $yr^{-2}$ for $N_2O$. The deceleration reversed the trend of ozone from an increase of ~ 0.8 ppb $yr^{-1}$ to a decrease of ~ 0.3 ppb $yr^{-1}$ over the 30-year data record, while the acceleration increased the trend of $N_2O$ from ~ 0.9 ppb $yr^{-1}$ to ~ 1.1 ppb $yr^{-1}$ over the 23-year record. The statistically significant value of the cubic term (i.e., the positive value of the $d$ parameter) indicates that the acceleration of the $N_2O$ rate of increase was not uniform; the rate increased more slowly near the middle (minimum ~2002) than at the beginning and end of the data record. In contrast, no statistically significant change can be discerned in the deceleration of the trend derived from the ozone data record. A caution to the above discussion should be noted – Eq. 5 is obtained by differentiation of an equation, whose parameters are derived from fits to data sets that have significant unexplained variability. As noted previously, Eq. 5 is not based on a physical model; hence, the above discussion regarding the rate of change must be considered cautiously, as discussed in Section 4.



## 3.8 Sources of Variance of data sets

The squares of the standard deviations of the original data sets that are annotated in Figures 1 and 6a give the total variance in the original data, and the square of the RMSD values that are annotated for all of the illustrated fits provide an approximate measure of the variance remaining in the data set after accounting for the average long-term change and/or the

seasonal cycle. Table 3 summarizes the fraction of the original variance of the monthly mean time series due to the average long-term changes and seasonal cycle. Despite the obvious differences of the data records in Figures 1 and 6a, the total variance per month of the ozone and $N_2O$ data sets are similar (36 and 29 $ppb^2$, respectively); however the source of that variance is quite different. The average long-term change accounts for only 19% of the ozone variance, but 99.7% of the $N_2O$ variance, while the seasonal cycle accounts for 58% and 0.19% of the ozone and $N_2O$ variance. The residuals thus

account for the remaining 23% and 0.12% of the variance; these residuals represent systematic interannual variability, i.e. the lower frequency "macroweather" regime of Bowdalo et al. (2016), and any other effects leading to variability in the data record (including any measurement errors or analysis biases).

## 4 Discussion and Conclusions

The analysis approach presented in this work derives a limited set of parameter values that defines a mathematical

representation of the statistically significant features of the mean long-term changes and seasonal cycles of the concentrations of trace tropospheric species. The results for the two example data sets (baseline concentrations of ozone and $N_2O$ at Mace Head Ireland) selected to illustrate the analysis show that no more than the seven or eight parameter values included in Table 2 are needed for this mathematical representation. Three or four parameters (the coefficients of the polynomials given by the first three or four terms of Eq. 1) quantify the long-term changes, including the absolute

concentration in the reference year 2000, and four parameters (the amplitude and phase of the two harmonic terms of Eq. 3) quantify the seasonal cycle. These parameters provide a robust, minimum set of parameters that capture the statistically significant information in the observational data set regarding these temporal variations. These parameters can serve as metrics for quantitative comparisons of long-term changes and seasonal cycles between different locations or for different species, and can serve as a basis for evaluation of model simulations atmospheric concentration variations.

In the quantification of mean long-term concentration changes, the method presented provides statistically significant information not given by linear trend analysis, an approach often employed to quantify long-term trends. Linear analysis does provide a quantification of the average annual rate of change of a species' concentration over the time span of the measurement record, but does not generally provide information about any statistically significant changes in the rate of concentration change (i.e., acceleration or deceleration of the rate of concentration increase or decrease) within the data

record. For baseline ozone concentrations, such changes of rate have been identified as quite important as shown here in Figure 1, and have been quantified in earlier work (Logan et al., 2012; Parrish et al., 2012; 2014; 2017; Derwent et al.,



2018a); these analyses show an increase early in the data record that slows, with concentrations reaching maxima, followed by decreases in the latter part of the record. In such cases, Eq. 2 provides an estimate of the year when the maximum concentration was reached. The Mace Head $N_2O$ record gives a contrasting result, with a statistically significant acceleration of the rate of increase throughout the data record, which is in agreement with an independent analysis of $N_2O$ trends, which

also identifies a significant acceleration of similar magnitude (Rona Thompson, private communication, 2018).

In the quantification of average seasonal cycles, the method presented here provides less detailed seasonal cycles than are sometimes derived in analyses of average seasonal cycles. Here it is shown that only four statistically significant, independent pieces of information are needed to quantify the mean seasonal cycles in the example data sets. Published studies often represent the seasonal cycle as twelve monthly means; such an approach implicitly assumes that there are

twelve statistically significant, independent pieces of information available from the mean seasonal cycle. At least for the example data sets examined in this and earlier work, (e.g., Parrish et al., 2016) the mean seasonal cycles are well described with no more than 4 independent pieces of information (i.e., independent parameter values) that can be extracted from the analysis. Including twelve monthly means in the seasonal cycle representation adds statistically insignificant variability to the results, and thus over fits to the available data, preventing a clear analysis of the statistical uncertainty of those results.

For the greatest statistical significance of the description of the seasonal cycle, we recommend a harmonic analysis that includes only significant terms, as exemplified in the method presented in this work. However, it is possible that other data sets from different locations may warrant more or fewer terms.

The analysis approach presented here is based on non-linear regression fits to Eq. 4, which assumes a number of properties about the behavior of the data, including that the data behave in a smooth manner, that the long-term change is independent

of season, and that the seasonal cycle is stable over the data record. Should any of these assumptions fail, or additional information be desired, the equation could be modified appropriately. The analysis derived a minimum set of statistically significant parameters that capture as much statistically significant information as possible from the original data sets, while avoiding over fitting the data. However no physical model underlies Eq. 4, so physical interpretation of the parameter values and extrapolation of the functional fit must be done only very cautiously. For example, the fit to the long-term changes in the

Mace Head ozone record indicates that maximum ozone concentrations occurred within 2.2 years of 2008.7, i.e. within ~26 months of September 1, 2008, and that after that maximum, ozone concentrations have been decreasing. The question arises as to whether the maximum and the subsequent decrease are physically real, or are simply mathematical implications of the three-term polynomial utilized to fit the data. Supplementary trend analyses indicate that 1) there has been no statistically significant trend after the year 2000 (average trend = -0.015 ± 0.070 ppb yr$^{-1}$), so it is at least clear that the positive trend in

the early years of the data record has ended, and 2) that any decrease after the derived maximum in the year 2008.7 is not statistically significant. Thus, answering the above question requires additional information that perhaps may come from additional years of data collected at Mace Head, or analysis of other ozone data sets that can be considered to reflect the same physical driving forces as those at Mace Head.



*Competing interests.* The authors declare that they have no conflicts of interest.

*Acknowledgements.* The authors gratefully acknowledge the cooperation and efforts of the operators of the Mace
Head Atmospheric Research Station and their support staff. The research facilities at Mace Head, Ireland were generously
provided by the School of Physics, National University of Ireland, Galway. Betsy Weatherhead of Jupiter, Rona Thompson
of NILU and Alistair Manning of the UK Met Office provided helpful discussions. The ozone data are available from
Derwent et al. (2018a) and the $N_2O$ data are available from the AGAGE, CDIAC, and WDCGG websites as described in
Section 2.

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

**Table 1. Parameter values of fits of long-term change in Mace Head, Ireland ozone data to Eq. 1.**

| Data fit | $a$ (ppb) | $b$ (ppb yr$^{-1}$) | $c$ (x10$^{-3}$) (ppb yr$^{-2}$) | RMSD (ppb) | $d$ (x10$^{-4}$) (ppb yr$^{-3}$) | year$_{max}$ |
|---|---|---|---|---|---|---|
| Monthly means | 39.8 ± 0.8 | 0.35 ± 0.08 | -20 ± 6 | 5.4 | 5 ± 11 | 2008.8 ± 4.2 |
| Annual means | 39.8 ± 0.7 | 0.31 ± 0.07 | -17 ± 8 | 1.3 | -4 ± 11 | 2008.9 ± 4.5 |
| Combined regression | 39.8 ± 0.4 | 0.34 ± 0.04 | -20 ± 4 | 2.9 | --- | 2008.7 ± 2.2 |

**Table 2. Parameter values of fits of long-term change and seasonal cycle in Mace Head, Ireland data to Eq. 4.**

| Data set | $a$ (ppb) | $b$ (ppb yr$^{-1}$) | $c$ (x10$^{-3}$) (ppb yr$^{-2}$) | $d$ (x10$^{-4}$) (ppb yr$^{-3}$) | $A_1$ (ppb) | $\phi_1$ (rad) | $A_2$ (ppb) | $\phi_2$ (rad) | RMSD (ppb) |
|---|---|---|---|---|---|---|---|---|---|
| Ozone | 39.8 ± 0.6 | 0.35 ± 0.06 | -20 ± 6 | --- | 5.7 ± 0.6 | 0.52 ± 0.10 | 3.1 ± 0.6 | -2.37 ± 0.19 | 2.9 |
| N$_2$O quad | 315.9 ± 0.2 | 0.76 ± 0.04 | +5.1 ± 3.1 | --- | 0.31 ± 0.17 | 0.48 ± 0.55 | 0.10 ± 0.17 | -3.42 ± 1.67 | 0.23 |
| N$_2$O cubic | 316.0 ± 0.1 | 0.76 ± 0.02 | -4.7 ± 5.1 | +6.0 ± 2.9 | 0.31 ± 0.09 | 0.48 ± 0.29 | 0.10 ± 0.09 | -3.39 ± 0.86 | 0.18 |

**Table 3. Sources of variance in Mace Head, Ireland data sets.**

| Data set | Total variance (ppb$^2$/month) | Long-term trend contribution (%) | Seasonal cycle contribution (%) | Residual (%) |
|---|---|---|---|---|
| Ozone | 36.1 | 19.2 | 58.0 | 22.7 |
| N$_2$O quad | 28.2 | 99.6 | 0.19 | 0.18 |
| N$_2$O cubic | 28.2 | 99.7 | 0.19 | 0.12 |





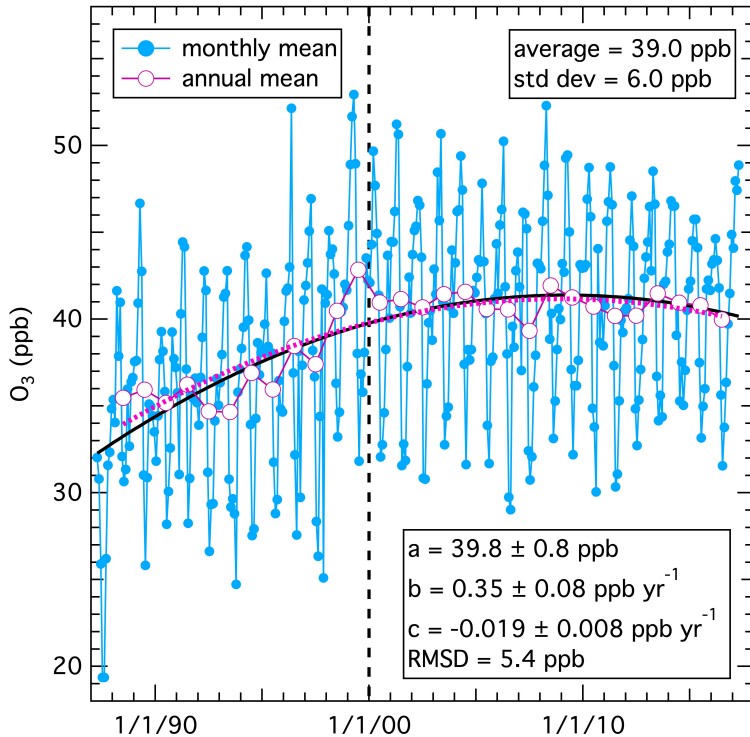

**Figure 1. Fits of long-term change in mean baseline tropospheric ozone measured at Mace Head, Ireland. Monthly means are from Appendix A of Derwent et al. (2018a), from which the annual means were calculated. The solid black and dotted violet curves are non-linear regression fits of the first three terms of Eq. 1 to the monthly and annual means, respectively.**





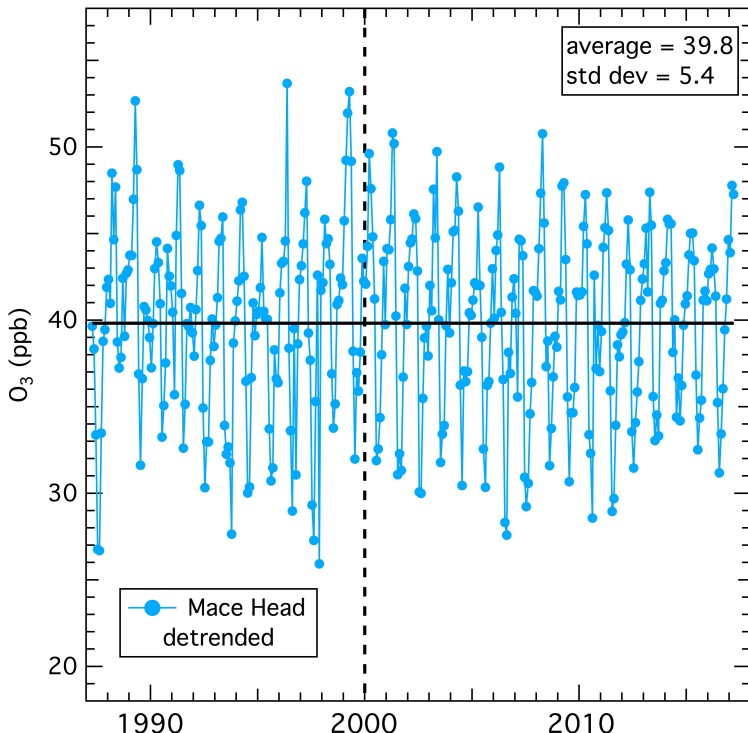

**Figure 2. Detrended monthly mean baseline ozone concentrations from Mace Head derived from the data of Figure 1. The solid black line indicates the mean of the data, equal to the *a* parameter from the fit in Figure 1.**


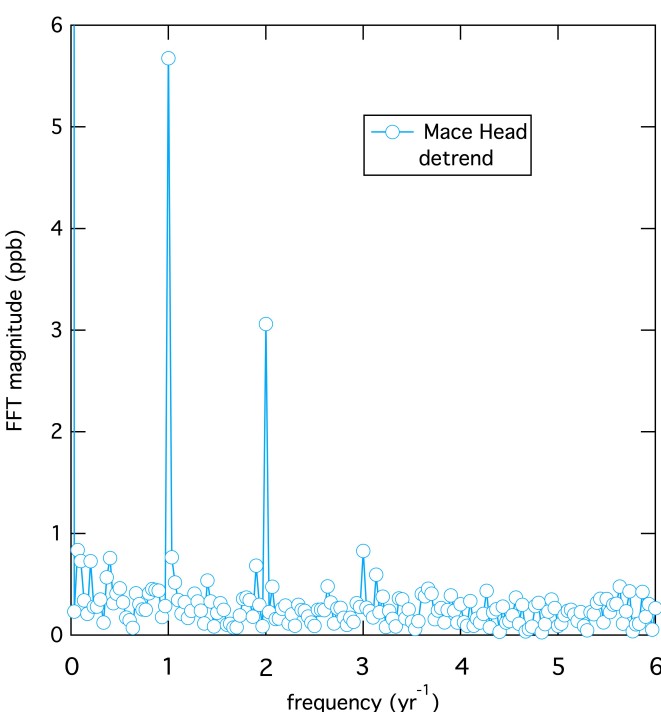

**Figure 3. Results of the Fourier transform of the detrended monthly mean ozone concentrations plotted in Figure 2.**

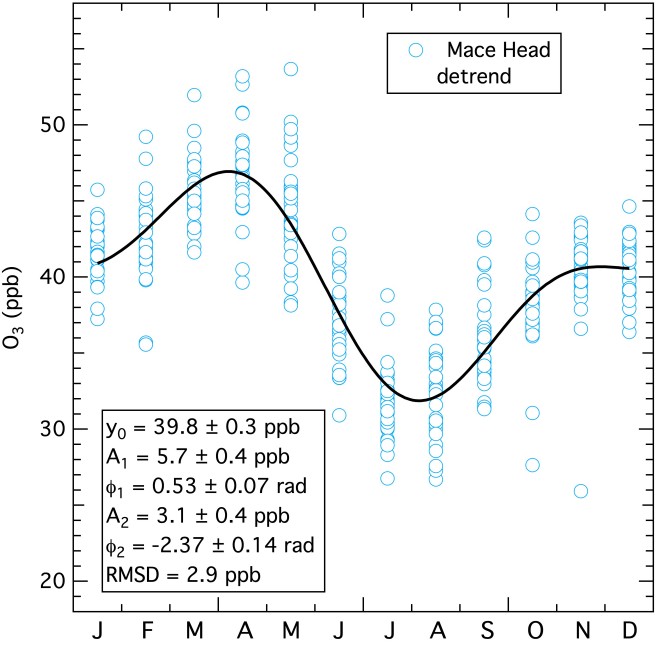

$y_0 = 39.8 \pm 0.3$ ppb
$A_1 = 5.7 \pm 0.4$ ppb
$\phi_1 = 0.53 \pm 0.07$ rad
$A_2 = 3.1 \pm 0.4$ ppb
$\phi_2 = -2.37 \pm 0.14$ rad
RMSD = 2.9 ppb

5 **Figure 4. Results of the fit of Eq. 3 to the detrended monthly mean concentrations from Figure 2. The black curve is the non-linear regression fit to the plotted points.**



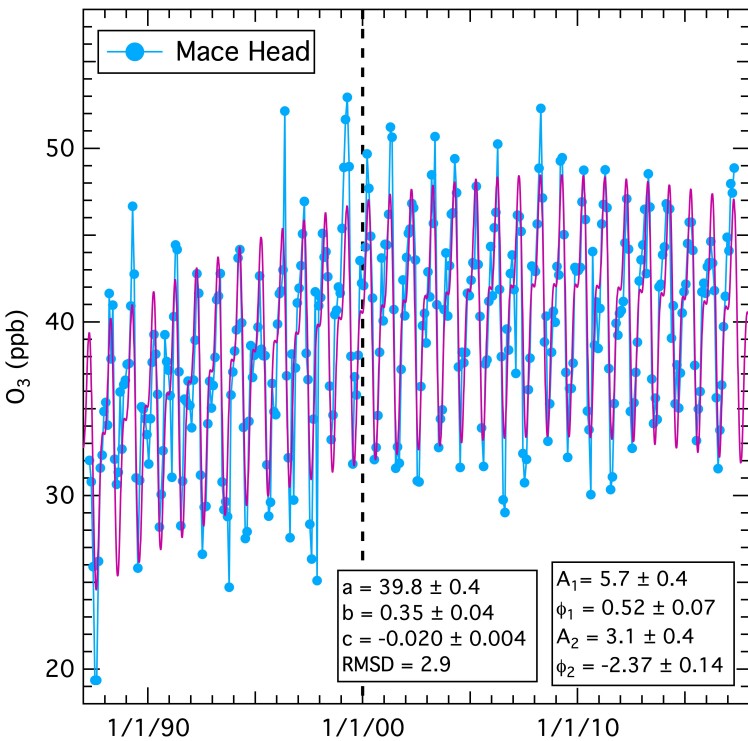

**Figure 5. Results of the non-linear regression fit of Eq. 4 (violet line) to the monthly mean concentrations from Figure 1. Table 2 gives the units of the parameters, and the confidence limits corrected for autocorrelation in the data set.**

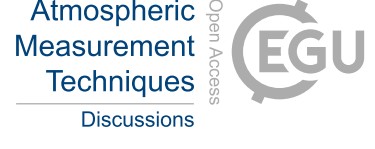

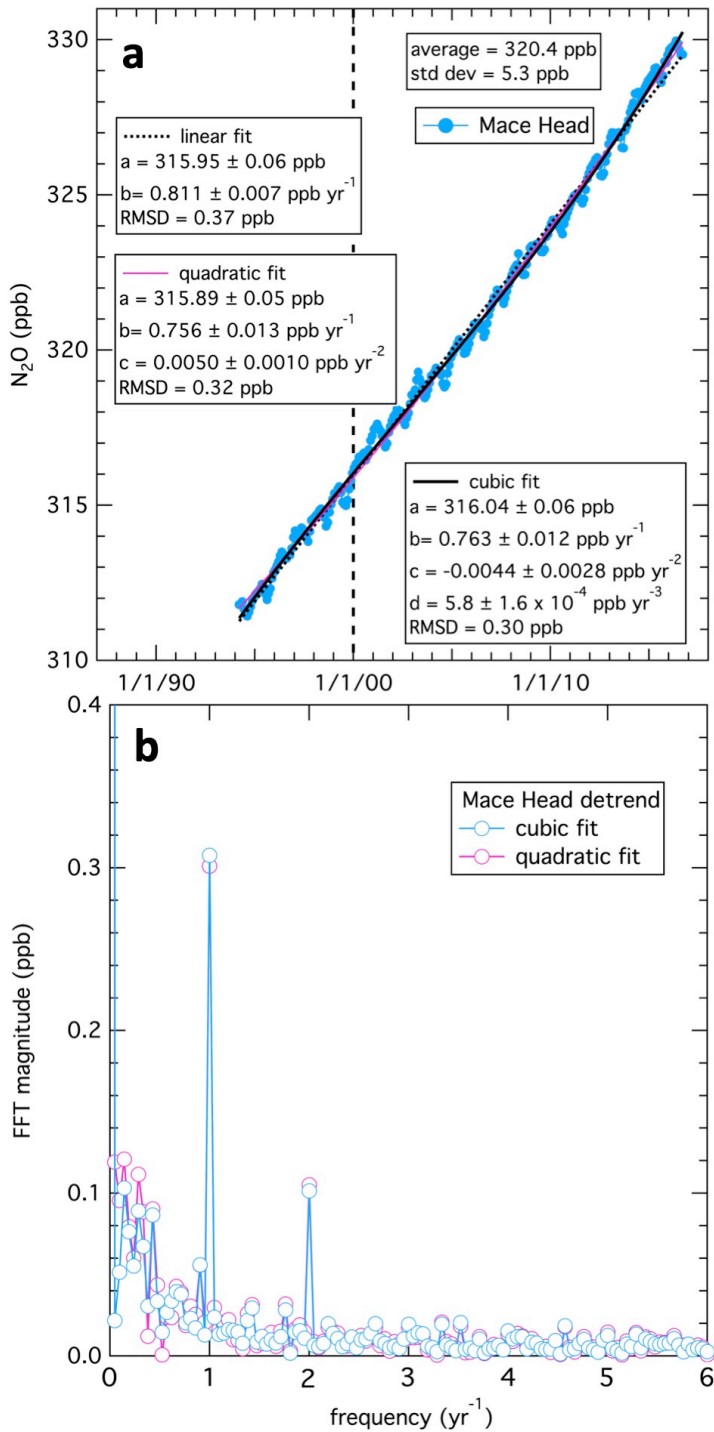





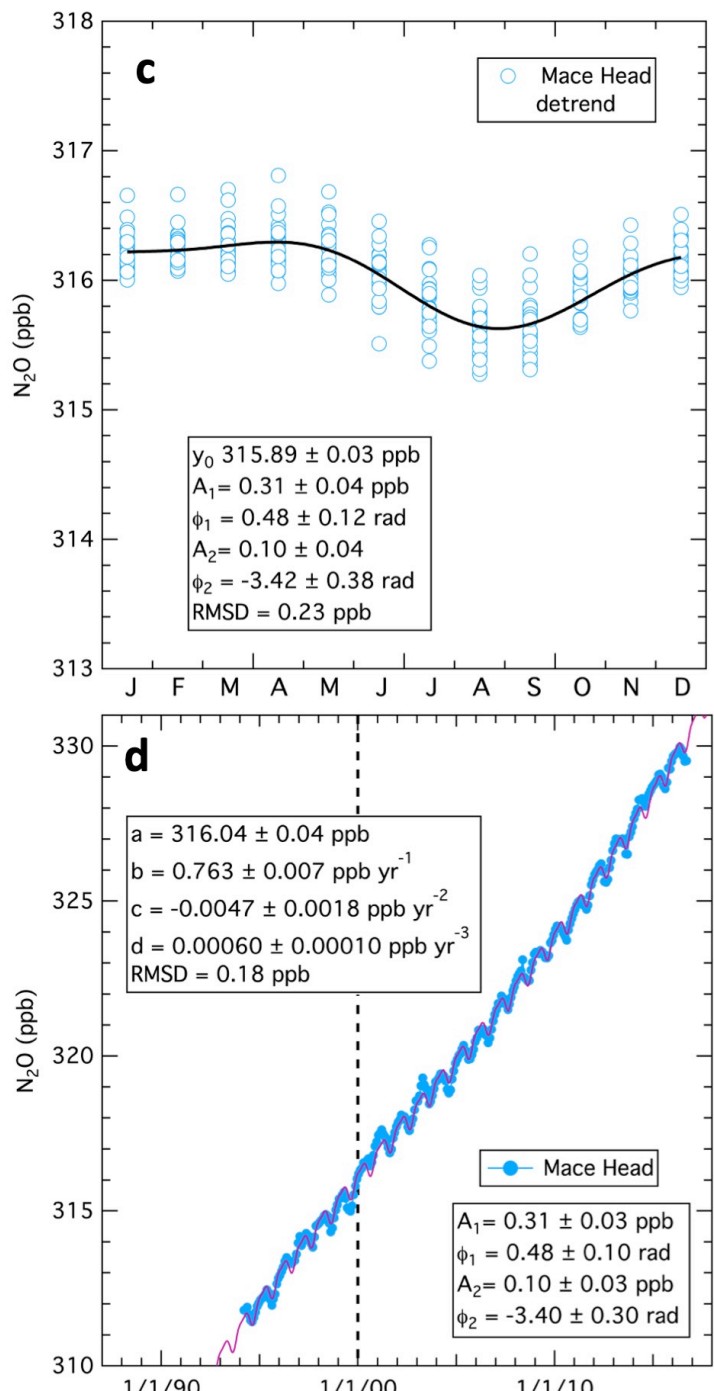

**Figure 6. Analysis results for a time series of monthly mean baseline nitrous oxide measured at Mace Head, Ireland. a) The three curves are fits of the monthly means to Eq. 1 with two terms (i.e., linear), three terms (i.e., quadratic), and four terms (i.e., cubic), with the derived parameters annotated. b) Results of the Fourier transform of the monthly mean concentrations detrended using the cubic (blue points) and quadratic fits (violet points). c) Results of the fit of Eq. 3 to the detrended monthly mean concentrations. The black curve is the non-linear regression of Eq. 3 to the plotted points. d) Results of the cubic fit (violet curve) of Eq. 4 to the detrended monthly mean concentrations. Table 2 gives the parameter values and the confidence limits corrected for autocorrelation in the data set.**





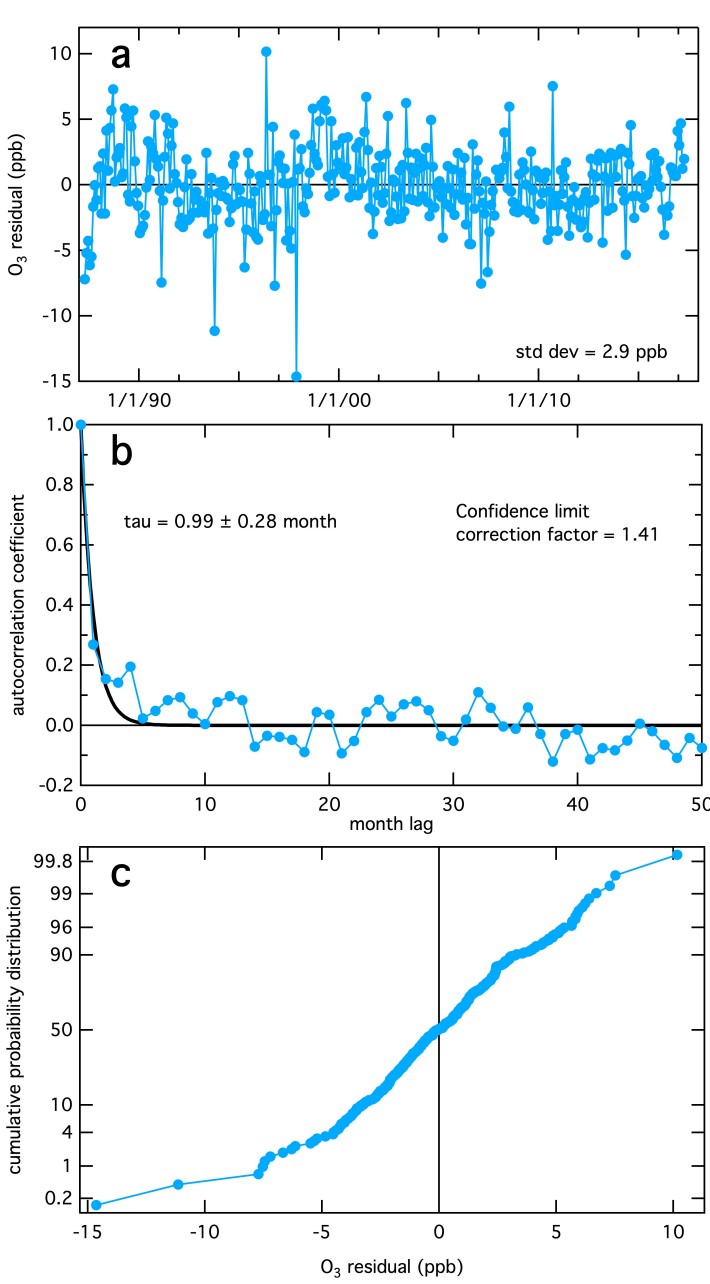

**Figure 7. Analysis of the deviations between the monthly mean baseline ozone concentrations and the fit of Eq. 4 to these means (i.e. the fit residuals) illustrated in Figure 5. a) Time series of the residuals. b) The time lag autocorrelation of the residuals. The fitted curve is an exponential decrease from unity at a lag of 0 month, with the time constant annotated. c) Cumulative probability distribution of the residuals, plotted on an ordinate scale that gives a linear fit for a Gaussian distribution.**



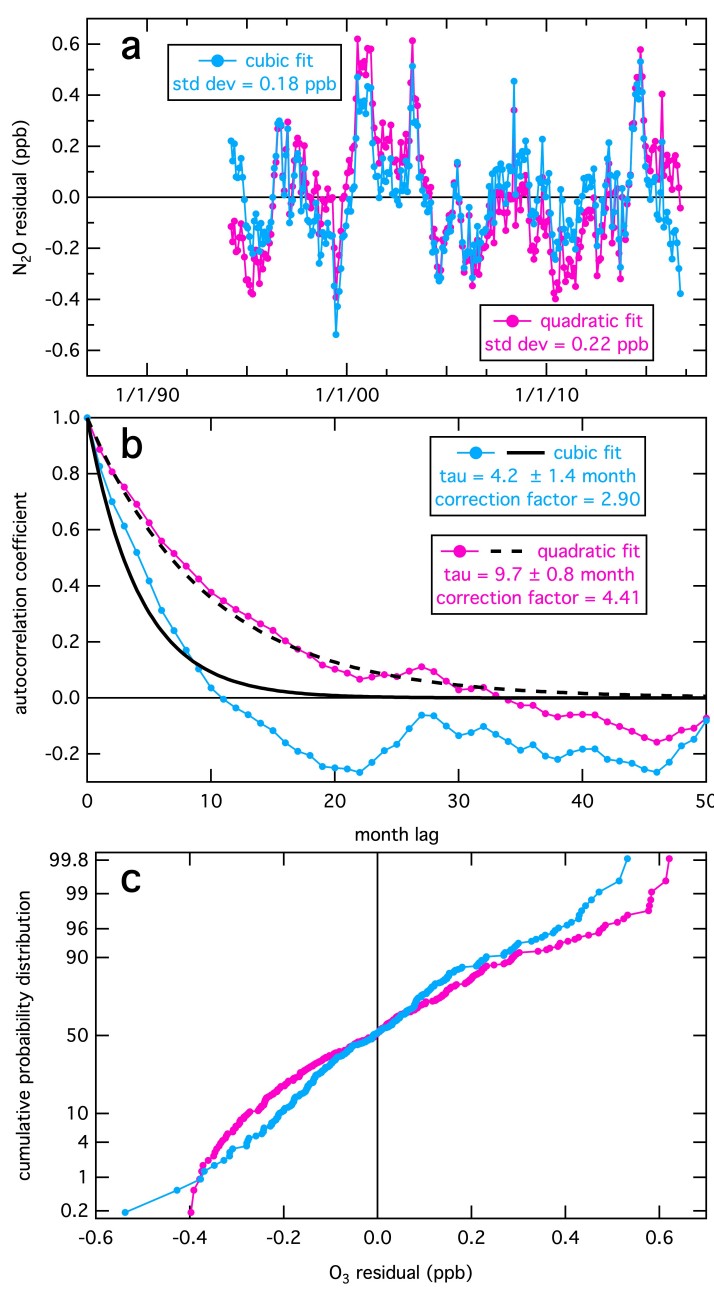

**Figure 8. Analysis of the fit residuals for the monthly mean baseline N$_2$O concentrations illustrated in Figure 6d (blue points). For comparison, the violet results show the analysis with a quadratic fit to the long-term changes. The annotations are similarly color-coded. The format of the figure is generally the same as that of Figure 7.**

