# Peer review of "Flexible approach for quantifying average long-term changes and seasonal cycles of tropospheric trace species"

_Atmospheric Measurement Techniques, 2019_

## Referee Comment (RC1) · Anonymous Referee #1 · 8 Apr 2019

Major comments:

The manuscript suggests using the power series and Fourier harmonics to represent long-term change and seasonal cycle of a time series, with an application to two gas species at Mace Head, Ireland. Overall, the manuscript is well written, the methodology is reasonable and the topic is of interest for the AMT journal. My only concern is the interpretation of the result (e.g. p5. l14.). The existence of a curvature in a time series does not automatically imply that there exists a meaningful or significant trend onward. It is extrapolating the fitted curve to an unobserved future based on a second order polynomial (due to it only allows one turnaround point), and is potentially misleading to

the reader.

An obvious example is that in the spirit of change point detection of a time series, one might reckon a significance of quadric term as a significance (or an existence) of one change point, and a non-significance of cubic term can be considered as no second change point occurred. The currently developed change point detection algorithms are designed for the detection of "mean" or "variance" change, and often fail to detect the change of the "trend", this is a big conceptual difference. Another example is that in the trend analysis of long-term of total ozone, many researchers use piecewise linear trend scheme to account for the ozone depletion since late 1970s and potentially recovery in late 1990s, but until now the latter trend for total ozone is yet to be significant in many literatures. Therefore the power series fit should not be over-interpreted as long-term changes, at least it is not representative for the trend after the maximum/minimum have reached (as stated in p10, l15 & l25, and in the conclusion the authors mention that trends after 200 and 2008 are not significant).

Minor comments:

P3. L30.: This statement only holds true if temporal sampling scheme does not introduce any bias to the monthly means. For example in the historical period some of the monthly means from ozonesonde might have only averaged from a very few profiles, it is then hardly be representative to the monthly means if there was an over- or under-sampling issue. The authors should clarify this point or be conservative to the statement.

P10. L21.: The term "robust" has a very specific meaning for the median or quantile regression in the statistics. The power series fit is still depended on the mean value and the ordinary least square, it is not allowed any breakdown point in the data, so it should not consider to be a robust approach.

Figs 7 & 8: The plot legends have labeled "tau" and "correlation factor", but these terms are not defined or mentioned in the main text.

[Figure]

---

## Author Comment (AC1) · 12 Apr 2019

The referee's comments regarding this paper are very helpful. In particular, the referee's concern regarding the interpretation of the curvature in a time series (page 5, line 14) is well founded. The curvature indeed does not automatically imply that there exists a meaningful or significant trend onward. And it is certainly true that extrapolating the fitted curve to an unobserved future based on a second order polynomial (or any other polynomial fit) is not only potentially, but almost certainly, misleading as polynomials generally diverge to large negative or positive values outside the data range used to derive the polynomial itself. As the referee notes, we do discuss this concern

in the Discussion and Conclusions Section. In the revised paper, we also now discuss this concern in the Analysis Approach Section as well; the original paragraph on page 5, lines 13-20 has been replaced with the following paragraph:

"The long-term fit to the Mace Head data finds a statistically significant, negative value for c, with ozone concentrations increasing early in the data record. The polynomial fit reaches a maximum, and then decreases later in the record. When three terms are included, Equation 2 allows the calculation of the year that the maximum of the fit was reached, year$_{max}$:

year$_{max}$ = -b/2c + 2000. (2)

The year$_{max}$ calculated from Eq. 2 is included in Table 1, which is within the time period of the observational record. The physical interpretation of the maximum derived from the fit, and any extrapolation to a maximum year outside the observational time period, would depend on the scientific understanding of the factors driving the concentration changes. As discussed later, the apparent decrease after the derived maximum of the fit in Figure 1 is not statistically significant, and the existence of a physical maximum of Mace Head ozone concentrations remains an open question. Extrapolation of fits derived from Equation (1) is likely misleading, since polynomials generally diverge to large negative or positive values when extended outside the data range used to derive the polynomial itself."

Minor comments by referee (in italics) and our response (in plain text):

*1) P3. L30.: This statement only holds true if temporal sampling scheme does not introduce any bias to the monthly means. For example in the historical period some of the monthly means from ozonesonde might have only averaged from a very few profiles, it is then hardly be representative to the monthly means if there was an over- or under-sampling issue. The authors should clarify this point or be conservative to the statement.*

A qualifying phrase has been added to the statement in question; it has been revised to read: "As long as the temporal sampling scheme does not introduce any bias to monthly means (e.g., through sparse sampling such that the data are not fully representative of the actual monthly means), restricting the analysis to monthly mean data rather than working with higher frequency data does not reduce the statistically significant information regarding the average long-term trends or seasonal cycles."

*2) P10. L21.: The term "robust" has a very specific meaning for the median or quantile regression in the statistics. The power series fit is still depended on the mean value and the ordinary least square, it is not allowed any breakdown point in the data, so it should not consider to be a robust approach.*

Thank you for this correction. The term "robust" has been removed from this sentence.

*3) Figs 7 8: The plot legends have labeled "tau" and "correlation factor", but these terms are not defined or mentioned in the main text.*

The main text did define "tau" (page 9, line 1). The plot legends have labeled a "correction factor" that was not explicitly defined. In the revised paper, this factor is now defined in Section 3.6: "Thus for the two example data sets discussed here, the confidence limits annotated in Figures 1 and 4-6 and included in Table 1 must be increased by a "correction factor" of 1.4 and 2.9 for ozone and $N_2O$ (for cubic long-term change fit), respectively; Table 2 gives the corrected confidence limits for the final values derived for the seven or eight parameters."

---

## Referee Comment (RC2) · Anonymous Referee #2 · 20 May 2019

Comments

Parrish et al. developed a useful approach to quantify the minimum set of parameters that mathematically describes the mean long-term evolution and seasonal cycle of an atmospheric trace species' concentrations. This is a continuous effort of the subjects to characterize the trend of the atmospheric trace compounds in the troposphere so that human response may be warned at appropriate moment. The major statistical tools implemented in this study is the power series fit for the annual trend and the Frouier Transform for the monthly change. The paper is within the scope of AMT and I have the following comments for the authors to consider before publication.

1. The power series fit as denoted by Equation 1 is certainly more useful compared with the normally used linear fit method. Nevertheless, could the authors describe the possible physical meaning of the changes subjected to tˆ2, tˆ3, tˆ4, . . .? In short, why power series fit? To fit the nonlinear trend, we could also try other function forms to describe the atmospheric oscillation. For example, if we think the ozone change may partly or largely be driven by the temperature, we may think a form of equation to describe the temperature change with the time.

2. The information of Figure 2 is quite limited which I think can be removed from the main text.

3. The Frouier Transform for the detrended monthly change of ozone concentrations shown in Figure 3 is very interesting but the information from Figure 4 is basically the same with a different view angle. I suggest to merge the two figures as one figure and assigned with two panels.

4. The section - 'The rate of change of the concentrations' is certainly very interesting, as the rate of change can be derived consequently as a differentiate of equation 4. Nevertheless, as long as the physical meaning is not clear, I suggest to remove this part or to add more discussions on this part to abstract the possible meanings of the phenomenological analysis with its theoretical background.

Technical comments: The time axis 1/1/99, 1/1/00, 1/1/10 better changed to be 1/1/1999, 1/1/2000, and 1/1/2010 or 1999, 2000, and 2010

The legend of Figure 1.: the unit of parameter c, should it be ppb yr ˆ{-2} Figure 5: it would be helpful to add a description of the violet line in the figure legend

---

## Author Comment (AC2) · 30 May 2019

We are grateful for the referee's careful reading and helpful comments. Following are the comments by the referee (*in italics*) and our response (in plain text):

*1. The power series fit as denoted by Equation 1 is certainly more useful compared with the normally used linear fit method. Nevertheless, could the authors describe the possible physical meaning of the changes subjected to $t^2$, $t^3$, $t^4$, $: : :$? In short, why power series fit? To fit the nonlinear trend, we could also try other function forms to describe the atmospheric oscillation. For example, if we think the ozone change may partly or largely be driven by the temperature, we may think a form of equation to*

*describe the temperature change with the time.*

This is a very important comment, and applies to the use of Fourier series to describe the seasonal cycle, as well as to the power series to describe the long-term changes. In our original submission we briefly discussed this issue in the final paragraph of the Introduction.

From a formal mathematical perspective, these series simply provide a means to quantify the long-term changes and seasonal cycle in a time series of measurements with the fewest possible statistically significant parameters. There is no implication that the individual terms have a physical cause, so one cannot necessarily attribute physical significance to any of the terms. Both series are mathematical representations of the respective systematic temporal variations; their utility arises because these mathematical representations are flexible enough to capture the statistically significant temporal variations in the time series. Bowdalo et al. [2016] discuss this issue for Fourier series quantification of temporal variability, including higher frequency temporal variations.

However, this formal perspective does not necessarily imply that one or more terms do not have an underlying physical cause. For the Fourier series, the fundamental is attributed to the yearly seasonal cycle, and indeed the form of Equation 3 is designed to facilitate that attribution (i.e., $\chi$ spans one year's time period in radians from 0 to $2\pi$). However, in the case of ozone in the marine boundary layer, Parrish et al., [2016] identify a physical process as the cause of the statistically significant second harmonic - the second harmonic of the photolysis rate of ozone, i.e., j($O^1$D), which causes a second harmonic in the loss rate of ozone. However this identification required information and analysis beyond that in the time series of concentration measurements.

If we have reason to believe that one or more other factors drive some of the temporal variability in a time series of concentration measurements (e.g. ozone change driven by temperature change) we could certainly fit a functional form that represents those causes through consideration of additional physical variables. However, that is beyond

the scope of the present work, as our goal is to develop a flexible analysis approach that does not require physical information beyond the time series of the concentration measurements.

Some of this discussion has been added to the final paragraph of the Introduction in the revised manuscript: "Without an underlying physical model, care must be exercised in the interpretation of the derived parameter values, and in the attribution of a physical cause to any of the terms in the series. Parrish et al. [2016] do present evidence of a direct physical cause of the statistically significant second harmonic of the seasonal cycle of ozone in the marine boundary layer (MBL) by showing that the photolysis rate of ozone, i.e. $j(O^1D)$, which drives the loss of ozone in the MBL, also has a second harmonic of opposite phase to that of ozone's seasonal cycle. However this identification required information and analysis beyond that of the time series of concentration measurements alone."

*2. The information of Figure 2 is quite limited which I think can be removed from the main text.*

Thank you. Figure 2 has been removed, and the following figures renumbered.

*3. The Fourier Transform for the detrended monthly change of ozone concentrations shown in Figure 3 is very interesting but the information from Figure 4 is basically the same with a different view angle. I suggest to merge the two figures as one figure and assigned with two panels.*

Thank you. However, due to some unfortunate technical issues, combining Figures 3 and 4 would be quite difficult; they have been retained as separate, newly numbered Figures 2 and 3.

*4. The section - 'The rate of change of the concentrations' is certainly very interesting, as the rate of change can be derived consequently as a differentiate of equation 4. Nevertheless, as long as the physical meaning is not clear, I suggest to remove this*

*part or to add more discussions on this part to abstract the possible meanings of the phenomenological analysis with its theoretical background.*

We agree that this is an interesting section. However, as discussed in the response to Comment 1) above, we cannot in general attribute a physical cause to the rate of change of concentrations. We have retained this section, and added sentence to the revised manuscript discussing the utility of quantifying the rate of change of concentrations: "Acceleration or deceleration in the rates of change of a species may contain information regarding changes in the magnitude of sources or sinks of the species, and thus may lead to improved physical understanding of the processes that determine the observed atmospheric concentrations."

**Technical comments:**

*The time axis 1/1/99, 1/1/00, 1/1/10 better changed to be 1/1/1999, 1/1/2000, and 1/1/2010 or 1999, 2000, and 2010*

Thank you. The time axes on all relevant figures have been changed to 1999, 2000, and 2010.

*The legend of Figure 1.: the unit of parameter c, should it be ppb $yr^{-2}$*

Thank you. This typo has been corrected.

*Figure 5: it would be helpful to add a description of the violet line in the figure legend*

Actually, the violet line was already briefly described in the figure legend (now legend for Figure 4); no change has been made here.

**References**

Bowdalo, D.R., Evans, M.J., and Sofen, E.D.: Spectral analysis of atmospheric composition: application to surface ozone model–measurement comparisons, Atmos. Chem. Phys., 16, 8295-8308, doi:10.5194/acp-16-8295-2016, 2016.

Parrish, D.D., et al.: Seasonal cycles of $O_3$ in the marine boundary layer: Observation and model simulation comparisons, J. Geophys. Res. Atmos., 121, 538–557, doi:10.1002/2015JD024101, 2016.